# Contributions of mechanical loading and hormonal changes to eccentric hypertrophy during volume overload: A Bayesian analysis using logic-based network models

**Johane H. Bracamonte**[1], **Lionel Watkins**[2], **Betty Pat**[3,4], **Louis J. Dell'Italia**[3,4], **Jeffrey J. Saucerman**[2], **Jeffrey W. Holmes**[1,4,5]*

**1** Department of Biomedical Engineering, University of Alabama at Birmingham, Birmingham, Alabama, United States of America, **2** Department of Biomedical Engineering, University of Virginia, Charlottesville, Virginia, United States of America, **3** Birmingham Veterans Affairs Health Care System, Birmingham, Alabama, United States of America, **4** Division of Cardiovascular Disease, Heersink School of Medicine, University of Alabama at Birmingham, Birmingham, Alabama, United States of America, **5** Division of Cardiothoracic Surgery, Heersink School of Medicine, University of Alabama at Birmingham, Birmingham, Alabama, United States of America

* holmesjw@uab.edu

## Abstract

Primary mitral regurgitation (MR) is a pathology that alters mechanical loading on the left ventricle, triggers an array of compensatory neurohormonal responses, and induces a distinctive ventricular remodeling response known as eccentric hypertrophy. Drug therapies may alleviate symptoms, but only mitral valve repair or replacement can provide significant recovery of cardiac function and dimensions. Questions remain about the optimal timing of surgery, with 20% of patients developing systolic dysfunction post-operatively despite being treated according to the current guidelines. Thus, better understanding of the hypertrophic process in the setting of ventricular volume overload (VO) is needed to improve and better personalize the management of MR. To address this knowledge gap, we employ a Bayesian approach to combine data from 70 studies on experimental volume overload in dogs and rats and use it to calibrate a logic-based network model of hypertrophic signaling in myocytes. The calibrated model predicts that growth in experimental VO is mostly driven by the neurohormonal response, with an initial increase in myocardial tissue stretch being compensated by subsequent remodeling fairly early in the time course of VO. This observation contrasts with a common perception that volume-overload hypertrophy is driven primarily by increased myocyte strain. The model reproduces many aspects of 43 studies not used in its calibration, including infusion of individual hypertrophic agonists alone or in combination with various drugs commonly employed to treat heart failure, as well as administration of some of those drugs in the setting of experimental volume overload. We believe this represents a promising approach to using the known structure of an intracellular signaling network to

**Data availability statement:** All data used for model construction, calibration, and validation were obtained from published studies identified within the manuscript and its Supporting information files. The source code and collected experimental data required to reproduce the model presented here including all results and figures are available at the open GitHub repository at https://github.com/cardiacbiomechanicsgroup/MCMC_cardiomyocyte_VO_growth.

**Funding:** This work is supported by the American Heart Association postdoctoral fellowship grant 23POST1026645 (https://doi.org/10.58275/AHA.23POST1026645.pc.gr.161204) to J.H.B; the National Institutes of Health and National Heart, Lung, and Blood Institute (NIH/NHLBI) grants R01HL159945 to J.W.H, R01HL162925 to J.J.S, and P01 HL051952 and P50HL077100 to L.J.D; Department of Veteran Affairs for Merit Review grant 1CX000993-01 to L.J.D, and Department of Veteran Affairs John B. Barnwell Award to L.J.D. The funders had no role in study design, data collection and analysis, decision to publish, or preparation of the manuscript.

**Competing interests:** The authors have declared that no competing interests exist.

integrate information from multiple studies into quantitative predictions of the range of expected responses to potential interventions in the complex setting of cardiac hypertrophy driven by a combination of hormonal and mechanical factors.

## Author summary

Mitral valve regurgitation is a common heart disease in which a malfunctioning valve allows part of the blood pumped by the heart to flow in the wrong direction. This condition overloads the heart by making it pump more blood volume than normal; the heart temporarily adapts by growing in mass and volume, but if untreated the condition can ultimately lead to heart failure and death. The most effective treatment is to surgically repair the valve; however, in many patients heart function deteriorates even after a successful surgery. Many researchers have studied this condition by experimentally overloading the hearts of dogs and rats, producing large amounts of data on the resulting geometric, mechanical, and biologic changes. Yet it has been difficult to translate those studies into effective selection and timing of treatment in all patients. In this work we integrate experimental data reported from 70 research articles on experimental volume overload through a simple model of heart mechanics and a more complex model of the molecular signaling pathways inside heart cells. We use a statistical approach to calibrate the computational model, so that it can predict not only average responses but also the degree of expected uncertainty for each prediction. We then use the model to explore how the heart responds to different combinations of drugs and hormones, and potential treatments during overload.

## 1. Introduction

Mitral valve regurgitation affects around 5 million people in America, and about 2% of the general population, with prevalence steeply increasing in individuals over 50 years of age [1]. In primary mitral regurgitation (MR) the dysfunction of one or more components of the valvular apparatus allows part of the blood volume pumped by the left ventricle to flow back to the low-pressure atrial compartment, making the heart pump a larger than usual volume of blood against a lower-than-normal resistance. The unique loading conditions imposed by MR and the resulting neurohormonal responses induce a distinctive ventricular remodeling response known as eccentric hypertrophy, consisting of the lengthening of individual cardiomyocytes by addition of sarcomeres in series, leading to an organ-scale dilation of the left ventricle volume with little change in its wall thickness [2,3]. The neurohormonal response to volume overload is characterized by the activation of the sympathetic and renin-angiotensin systems, similar to other forms of cardiac overloading [4–6]. Drug therapies for heart failure due to primary MR alleviate symptoms and slow its progression, but only mitral valve repair can provide

significant recovery of cardiac function and dimensions [2,7]. If MR is severe enough or if it remains untreated for long enough, the condition can transition from a compensated asymptomatic stage into irreversible heart failure with systolic dysfunction, a condition where the heart is unable to supply sufficient cardiac output to the body [8,9]. This risk has led clinicians to operate earlier in the natural course of primary MR; yet 20% of patients still develop systolic dysfunction post-operatively despite being treated according to the current guidelines [10,11]. This fact highlights our incomplete understanding of eccentric hypertrophy due to primary MR and its transition into systolic dysfunction and heart failure. A better understanding of this process is needed to improve and better personalize the management of MR.

Most computational models of growth and remodeling during volume overload have focused on the role of myocyte overstretch in driving eccentric hypertrophy [12,13]. By contrast, molecular studies have shown reduced activity in stretch-sensitive myocyte signaling pathways during experimental volume overload, the opposite of what would be expected if remodeling is driven by stretch [14–18]. We hypothesized that this apparent paradox might stem from complex interactions between the hypertrophic signaling pathways triggered by stretching and those that respond to other hormones and growth factors known to be upregulated during volume overload.

Here, we employ a Bayesian approach to combine the wealth of available data on experimental volume overload in dogs and rats using a logic-based network model of hypertrophic signaling in myocytes, with the goal of better understanding the relative influence of multiple factors that influence eccentric hypertrophy. We synthesized data from 70 studies of experimental volume overload to estimate the relative influence of multiple input parameters for a network model of hypertrophic signaling in cardiomyocytes during volume overload, accounting for evolving levels of mechanical strain and circulating hormones such as norepinephrine (NE), angiotensin II (AngII), endothelin 1 (ET1), and atrial (ANP) and brain (BNP) natriuretic peptides. We then validated the ability of the calibrated model to reproduce features of volume overload not included in the calibration, as well as experimental responses to relevant independent experiments such as infusion of hormones that induce myocyte hypertrophy, alone or in combination with receptor blockers used clinically to treat heart failure, from 43 independent research articles.

The calibrated and validated model developed here represents a probabilistic, model-driven meta-analysis of a large body of data on volume-overload hypertrophy. Our analysis suggests that elevated levels of circulating hormones drive much of the hypertrophic response during late stages of experimental volume overload, whereas hormone-driven growth frequently reduces myocyte strain levels below baseline despite elevated left ventricular volumes. These results contrast with the assumption of most computational models that elevated myocyte stretch drives eccentric hypertrophy but agree with much of the available molecular and signaling data. The performance of the model in simulations of multiple independent experiments not used in its construction suggest that the approach presented here is promising for better representing the multifactorial complexity of a condition like volume overload, as well as the responses to drugs administered in this complex setting. However, these simulations also highlight that responses to some interventions such as β-adrenergic blockade will require a multiscale approach that considers both direct effects on hypertrophic signaling as well as indirect effects through changes in mechanics and hemodynamics.

## 2. Methods

### 2.1. Data collection

We reviewed and collected data from 37 research articles on experimental mitral regurgitation in dogs and 33 articles on experimental volume overload by aorto-caval shunt in rats. All data employed for our quantitative analysis were reported as a mean value and standard deviation, so we assumed a normal probability distribution function (PDF) for all measurement-derived variables. Any single observation in the form of a mean and standard deviation will be called a dataset in this work. For the estimation of myocardial stretch at tissue scale, we focused on canine experiments to avoid confounding effects of growth in body size common during experimental volume overload in rodents. We collected data

on changes in left ventricular mass (LVM), end-diastolic volume ($V_{ED}$), and free wall thickness ($h_{ED}$), as well as previously reported estimates of end-diastolic myofiber stretch in healthy dogs $\left(\lambda_{ED}^0\right)$.

Both experimental volume overload (VO) and naturally occurring MR in dogs trigger elevated circulating levels of multiple hormones relevant to hypertrophic signaling, including AngII, NE, ET1, ANP and BNP, all of which are included as inputs to the cardiomyocte signaling network model employed here. We collected all data on plasma or serum concentrations of these hormones reported in the reviewed articles. We also collected data from the same studies on activity and phosphorylation levels of intracellular signaling proteins represented as intermediate nodes in the cardiomyocyte signaling model. Specifically, we collected Western blotting data on focal adhesion kinase (FAK), Akt, ERK5, ERK12, ELK1, cGMP, p38, and JNK from myocardial tissue extractions collected at several stages of VO. Additionally, we collected data on the abundance of several proteins synthesized by myocytes in tissue samples extracted following chronic VO including SERCA, myosin heavy chain isoforms αMHC and βMHC, ANP and BNP [19]; the network model predicts expression of the corresponding genes as outputs. A detailed list of sources for all the collected quantitative data is summarized in Tables A and B in S1 Text.

## 2.2.  Integration of canine and rat experimental data

Plots of the experimental fold change of normalized left ventricular mass to body mass (LVM/BM) during volume overload showed very similar shapes for dogs and rats, but hypertrophy occurred much faster in rats [9]. When we fitted data from each species with an exponential function and normalized the time axis by the time constant of that fit, we found that data from both species aligned (Fig 1a, the full list of data sources is shown in Table A in S1 Text). We therefore normalized all time course data in this study by the fitted time constant for each species. This allowed the use of combined data from both animal models in our quantitative analysis.

## 2.3.  Time-varying hormonal input curves

We found that baselines or control values of relevant hormone concentrations in blood were consistent across animal sizes and species, suggesting a common homeostatic range of circulating concentrations for each hormone. In this work, we assume that fold-changes in concentration levels of those circulating hormones represented the intensity of the neurohormonal response and would trigger proportional changes in the hormone-receptor reaction input in the cardiomyocyte signaling model. The time-resolved data of serum concentrations were normalized to their corresponding baseline or control concentration and plotted as a function of characteristic growth time (t/$\tau$). We confirmed that data from both species followed similar trends and fitted the integrated experimental data with the simplest function (linear or exponential) that captured the temporal trends (Fig 1). Details of the fitting process and the specific functions employed for fitting are provided in S1 Text. The fitted, time-varying probability distribution functions (PDFs) for each hormone were used as inputs to the signaling network model in subsequent simulations of volume overload.

## 2.4.  Time-varying myoStrain input curve

In addition to the many hormonal changes discussed above, experimental volume overload alters left ventricular (LV) mechanics acutely, followed by further evolution due to the ongoing process of eccentric hypertrophy. The network model provides an input labeled myoStrain as a generic representation of the stretch-activated signaling pathways thought to be involved in transducing changes in myocyte mechanics into hypertrophy. Because a number of successful models of eccentric hypertrophy during VO have employed end-diastolic strain relative to an unloaded reference state as a growth stimulus [12,13], we assumed here that the myoStrain network input varies proportionally with end-diastolic strain in the LV. We computed a time-varying PDF for the fold change in end-diastolic strain using published data and a simple geometric model of a thin-walled sphere, as outlined below.

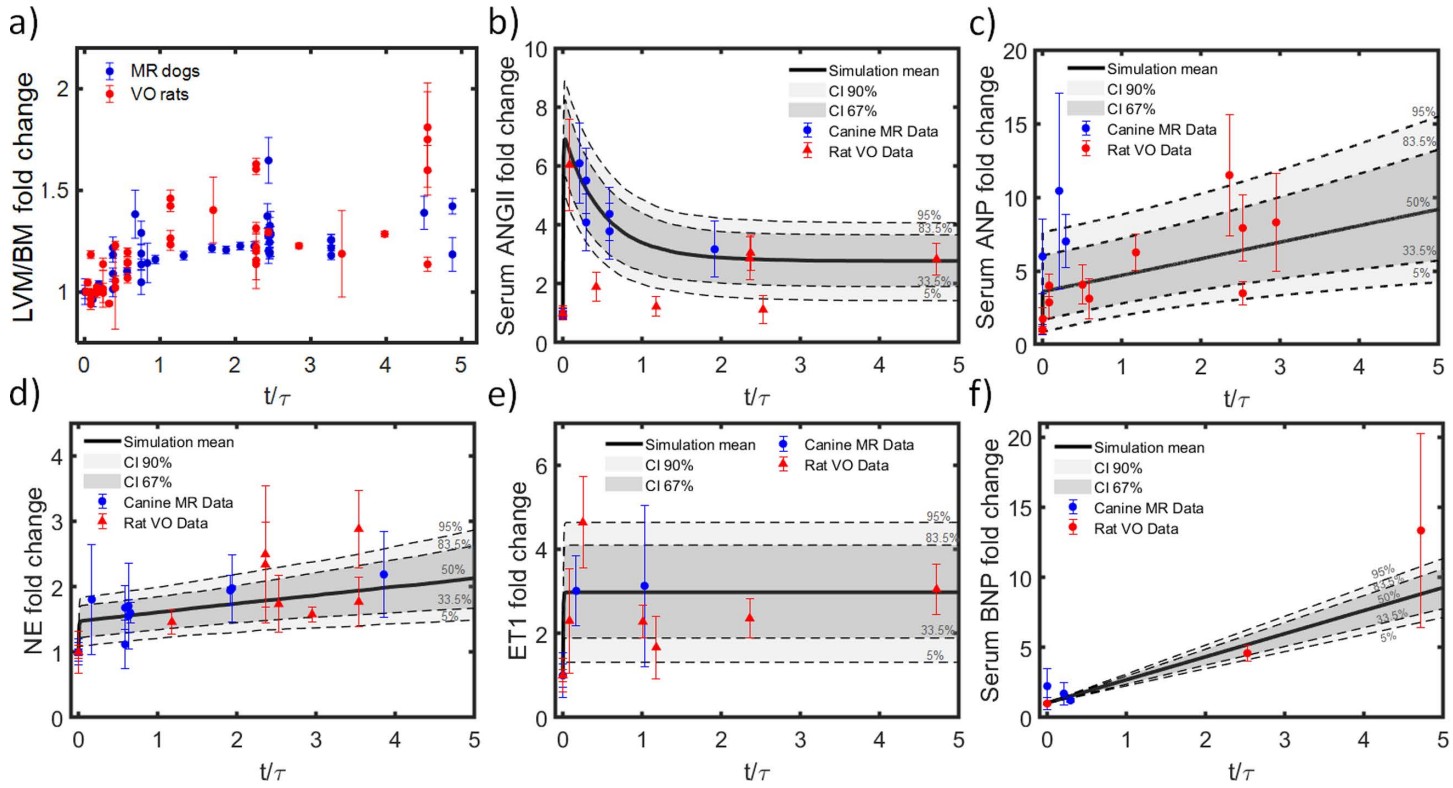

**Fig 1. Integration of experimental data from MR in dogs and VO in rats.** Each plot shows fold changes in one reported measure, plotted as a function of time normalized by the time constant of an exponential equation fitted to the LVM/BW data. a) LVM/BM data from sixteen articles (41 datasets) on experimental MR in dogs [15,16,20–33] and seventeen articles (45 datasets) on experimental VO in rats [34–50]. b) 6 measurements of serum angiotensin II (ANGII) from three articles on experimental MR in dogs [29,30,51] and 7 measurements from six articles on experimental VO in rats [38,41–43,52,53]. c) 3 measurements of serum atrial natriuretic peptide (ANP) levels from three studies in dogs [51,54–56] and 9 measurements from five studies in rats [39,53,57–59]. d) Norepinephrine (NE): 10 datasets from four MR studies in dogs [21,31,60,61] and 7 datasets from six studies in rats [39,44,48,52,59,62]. e) Endothelin 1 (ET1): two datasets from two studies in dogs [63,64] and two datasets from two studies in rats [36,42]. f) Brain natriuretic peptide (BNP): three datasets from three articles on experimental canine MR [51,54–56]; two datasets from two articles on experimental VO in rats [38,57]. Probability distribution functions (PDFs) are derived from fits to equations listed in Table C in S1 Text. PDFs are presented in shaded gray, median in solid line and quantiles in dashed lines.

### 2.4.1. Calculating end-diastolic strain relative to an unloaded state.

Let the circumference of a mechanically unloaded sphere (i.e., floating in saline without any applied pressure) be $2\pi r_0$, where r is the radius of the sphere and the subscript zero indicates the unloaded or reference state. Once inflated to end-diastole, the sphere's circumference is $2\pi r_{ED}$. The stretch ($\lambda$) around its circumference is defined as the ratio of loaded to unloaded length:

$$\lambda = \frac{2\pi r_{ED}}{2\pi r_0} = \frac{r_{ED}}{r_0}$$

(1)

In a spherical model of the LV, myocytes with any orientation parallel to the surface will also stretch by this same fraction. While an actual heart is never fully unloaded unless it is arrested and removed from the body, most mechanics models of growth use the unloaded state as a mathematical reference point because it is easy to define and doesn't change with beat-to-beat variations in hemodynamics.

Since many studies reviewed here tracked changes in volumes, we found it convenient to express stretch and strain in terms of volumes rather than radius. Recognizing that the volume of a sphere is proportional to $r^3$, we can rewrite the equation for stretch as:

$$\lambda = \frac{r_{ED}}{r_0} = \left( \frac{V_{ED}}{V_0} \right)^{1/3}$$

(2)

Finally, for large deformations measured relative to a fixed reference state, the most appropriate measure of strain is the Lagrangian strain [65]:

$$\varepsilon_f = \frac{1}{2} \left( \lambda^2 - 1 \right) = \frac{1}{2} \left( \left( \frac{V_{ED}}{V_0} \right)^{2/3} - 1 \right)$$

(3)

**2.4.2. Effect of growth on end-diastolic strain.** Returning to the unloaded sphere, if each myocyte grows longer by adding sarcomeres in series – as occurs during eccentric hypertrophy – then the unloaded radius $r_0$ and volume $V_0$ will be larger, and the stretch at a given end-diastolic volume will be smaller (Eq 3). In other words, growth changes the amount of stretch each myocyte experiences, which alters the stimulus for further growth. One of the key features of the model presented here is that it accounts for this critical feedback between growth and the hypertrophic signaling that produces it. Fig 2a illustrates this point graphically. In a spherical model of a normal heart, stretch increases as a nonlinear function of volume during simulated inflation. A modest amount of growth (10% volume increase) shifts the entire curve rightward, so that stretch is lower at a given volume, and inflation to a larger volume is required to achieve the same stretch.

A thin-walled sphere is a very simplistic representation of the geometry of the left ventricle, and a more sophisticated geometric model would produce a different quantitative relationship between volume and stretch. But that was not an important limitation for this study, for three reasons. First, the spherical model does remarkably well when compared to actual sarcomere lengths measured by Ross et al. in normal and hypertrophied hearts fixed at controlled volumes (Fig 2b), suggesting its use here is reasonable [66]. Second, the trends we predict here – an acute increase in end-diastolic strain followed by a gradual decline to baseline levels at 6 weeks – were experimentally verified by

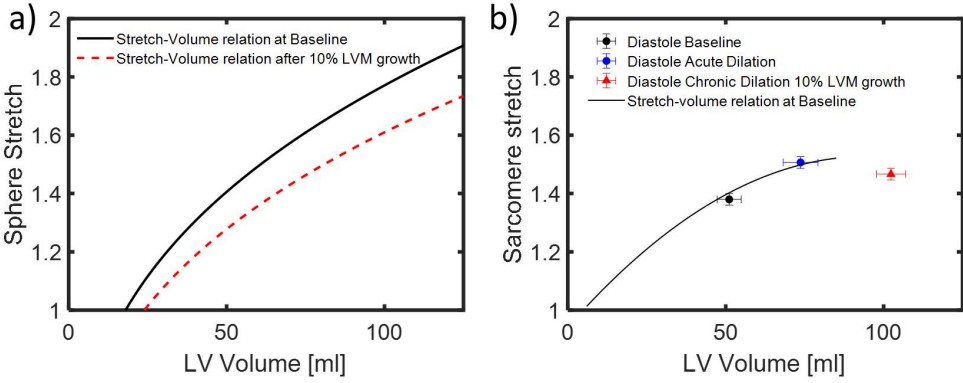

**Fig 2. Effects of inflation and growth on end-diastolic stretch relative to unloaded state in a spherical model and in hearts fixed at control volumes.** a) Spherical model predicts that stretch increases with inflation (black solid line), while eccentric hypertrophy shifts the stretch-volume curve downwards, to lower stretch at any given volume (red dashed line). b) Experimental measurements of sarcomere stretch in dog hearts fixed at controlled volumes similarly show increased stretch between physiological loading conditions (Baseline) and acute volume overload (Acute dilation), and a shift in the stretch-volume relationship following 10% growth in LVM during long-term volume overload. Figure generated based on data from Ross et al. [66].

Emery and Omens (1997) using markers attached to the LV surface [67]. Finally, it is important to note that the stretch and strain values we calculate from the spherical model (or any alternative model) are ultimately mapped onto a normalized input to the signaling network, and the relative influence of that input is calibrated against data (see below). Therefore, a different geometric model would just produce a different calibration; the overall behavior of the system would not be affected.

Consistent with the approach described above for integrating data on neurohormonal alterations, we estimated the time-varying PDF of strain by integrating the spherical model with data from sixteen articles reporting changes in LVM [15,16,20–33] and twenty-one articles reporting changes in $V_{ED}$ during experimental MR in dogs, as well as five articles reporting end-diastolic myocardial or sarcomere stretch relative to baseline in healthy dogs [66,68–71]. Details of this process are described in S1 Text.

**2.4.3. Mapping end-diastolic strain onto the network input myoStrain.** The mechanical input to the cardiomyocyte network model (myoStrain) is a normalized variable ranging from 0 to 1 (Section 2.5). However, in practice, increasing this input from its default baseline of 0.06 to just 0.2 produces the maximum possible CellArea output (CellArea=1) in the published version of the model employed here [72]. Thus, we sought to design a function that maps the range of expected values for organ-scale end-diastolic strain at baseline or during volume overload into myoStrain input values within the 0-0.2 working range. We also wanted the mapping function to produce a zero myoStrain input when organ-scale strain was zero (i.e., when the LV is in a completely unloaded state). We decided to use an exponential mapping function of the form:

$$ w^j_{myoStrain} = C_{myo} \left( e^{D_{myo}\ \varepsilon^i_f/\varepsilon^0_f} - 1 \right) $$

(4)

where superindex i refers to any given time, while superindex 0 is reserved for the baseline (pre-VO) state, $\varepsilon_f$ and $w_{myoStrain}$ are the maximum fiber strain and its corresponding mapped myoStrain, and $C_{myo}$ and $D_{myo}$ are mapping parameters to be fitted by a Markov Chain Monte Carlo (MCMC) algorithm (Section 2.6). To reduce the order of the MCMC parameter space, we fixed the baseline myoStrain weight $\left( w^0_{myoStrain} \right)$ at a single value for each MCMC run and repeated the process for $w^0_{myoStrain}$ values of 0.02, 0.04, 0.05 0.055, 0.06, 0.065, 0.07, 0.08, and 0.09. On each MCMC iteration we assign a random value to $C_{myo}$ and calculate $D_{myo}$ with Eq 4, the prescribed value of $w^0_{myoStrain}$, and reported baseline values of strain $\left( \varepsilon^0_f \right)$ [66,68–71].

### 2.5. Model of cardiomyocyte hypertrophic signaling pathways

We employed a published computational model of the hypertrophy signaling network that integrates many established pathways implicated in cardiac myocyte growth. The model consists of a logic-based network where the activity of each node follows a normalized Hill equation with possible activity values ranging from 0 to 1 [19,73]. The network consists of 106 nodes and 192 reactions. The nodes represent chemical and mechanical inputs, intracellular signaling proteins, transcription factors, and genes relevant to hypertrophy. The model has been used previously in the study of ventricular hypertrophy and was recently optimized in the context of β-adrenergic stimulation [72]. The set of network parameters is summarized in S1 Material. In Fig A in S1 Text we show a representation of the network model highlighting the nodes for which experimental data are available from experimental VO in rats or dogs.

The influence of a reaction on the downstream nodes is modulated by the weight parameter w, which was left at the default value for all nodes except the inputs AngII, ANP/BNP, ET1, NE, and stretch. The characteristic time constant governing the speed of changes in node activity was chosen as $0.005\tau$ for all intracellular reactions, $0.02\tau$ for output nodes reflecting gene expression, and $\tau$ for CellArea, where $\tau$ is the fitted time constant for the exponential rise in LV mass (LVM), as discussed above. The network model was solved with Netflux (https://github.com/saucermanlab/Netflux). More detail about the network model formulation and solution method can be found in [74].

## 2.6. Bayesian inference parameter estimation

All parameter estimations required in our data processing pipeline were performed within a Bayesian inference framework. The Bayesian inference tool utilized for this study was a standard Markov Chain Monte Carlo (MCMC) algorithm with Metropolis-Hasting selection criteria and Gibbs sampling to navigate the multiparametric space. Briefly, the algorithm iteratively solves a numerical model while randomly varying its input parameters over a predetermined probability distribution, known as the prior probability distribution function (prior PDF) of the parameter space. On each iteration, the likelihood of the model's outputs is evaluated against experimental data. If the likelihood of the outputs with the current parameter set is larger than the likelihood of the previous iteration, the parameter set is saved. If the outputs for the current parameter set are less likely, the decision on whether to save the current parameter set is made randomly. After sufficient iterations, the collection of saved parameter sets converges to a new probability distribution, or posterior PDF of the parameter space, which are associated with probability distributions for the model predictions [75,76].

In this study, each MCMC algorithm was applied in two stages, first assuming a uniform probability distribution of the parameters within their physiologically plausible limits for 10,000 iterations. The resulting posterior PDF was then used as the prior PDF for a second run of the MCMC algorithm for an additional 20,000 iterations, with a check to verify the convergence of the solutions every 5,000 iterations (Figs F and G in S1 Text). The MCMC algorithm was programmed in MATLAB.

**2.6.1. *Probability distribution of hypertrophy network input weights*.** The normalized time-varying curves of hormone concentrations and mechanical strain provide information on how these stimuli vary over time during volume overload, but not on their relative influence in driving hypertrophy. One advance of the current work over previous applications of this network model is that we allow the key hypertrophic stimuli to have different weights, and calibrate those weights using experimental data. In the cardiomyocyte signaling network model, the baseline weight of the hormone-receptor reaction determines its relative influence on the network, because the magnitude of the input at any given time is calculated as the product of its baseline weight $\left(w^0_{species}\right)$ and its fold-change from baseline at the current time (sections 2.3 and 2.4). We employed an MCMC to estimate the PDF of the baseline weights of hormone-receptor input reactions as follows. We first assumed a uniform prior PDF for the input weights of ANGII $\left(w^0_{AngII}\right)$, NE $\left(w^0_{NE}\right)$, and ET1 $\left(w^0_{ET1}\right)$ reactions. Sampling was constrained within the range for which the CellArea output is sensitive to those inputs. Specifically,

$$0.01 < w^0_{ANGII} < 0.15$$

$$0.01 < w^0_{NE} < 0.24$$

$$0.01 < w^0_{ET1} < 0.17$$

(5)

For the rest of the input reactions, we assign a single "background" reaction weight, sampled within the $0.01 < w_{background} < 0.4$ range. A preliminary study revealed that, within the range of interest, the input reaction weights of ANP and BNP to Guanylate Cyclase A (GCA) receptors have only marginal effects on predicted changes in CellArea; we therefore prescribed ANP and BNP the same background weight as the other inputs for which limited experimental data were available. We assigned null weight to the input reactions for the exogenous drugs phenylephrine and isoproterenol (ISO) except when simulating drug infusions.

On each step of the MCMC, the algorithm randomly samples the $w^0_{ANGII}$, $w^0_{NE}$, $w^0_{ET1}$, $w_{background}$, and $C_{myo}$ parameter space and randomly selects time-varying curves for each stimulus from their respective PDFs. The likelihood of each model run was evaluated against experimental data on FAK (7 datasets, 3 studies) [14,15,25], Akt (2 datasets, 2

studies) [15,77], ERK5 (1 dataset, 1 study) [78], ERK12 (7 datasets, 3 studies) [15,50,79], ELK1 (1 dataset, 1 study) [50], cGMP (4 datasets, 4 studies) [53,78,80,81], p38 (6 datasets, 4 studies) [15,78,79], and JNK (5 datasets, 2 studies) [15,79] activity and cardiomyocyte growth, CellArea in our model, (11 datasets, 9 studies) [16,21,23,28,29,34,82–84] from dog and rat experiments. We added a condition assigning larger likelihoods to parameter sets that produce a baseline CellArea activity near 0.5, in the most responsive region of the sigmoidal curve. Long-term experiments agree that LVM plateaus at a new level in chronic stages of VO [26]. We therefore assumed that continued growth at late time points and negative growth (reversal of hypertrophy) at any time point were very unlikely. After convergence of the MCMC, we filtered out these very unlikely solutions and recorded the posterior PDFs of the activity of the network nodes of interest.

### 2.7. Sensitivity analysis

We evaluated the sensitivity of network outputs (expression of ANP, BNP, αMHC, βMHC and SERCA) to the network inputs (ANGII, NE, ET1, ANP and BNP) by a standard correlation matrix based on statistical linear regression. The Pearson correlation coefficient (PCC) was calculated to quantify the parameter sensitivity. This method exploits the wealth of samples produced during the MCMC runs to yield sensitivity estimates that are meaningful within the expected range of network activity.

### 2.8. Validation

Volume overload is a complex process driven by multiple stimuli that trigger intersecting signaling cascades. We selected VO for the calibration of the hypertrophy network model to take advantage of the wealth of published data on how these different pathways respond over time. We calibrated the relative influence of the hormonal and mechanical stimuli by adjusting the weight of the network input reactions to match data on LVM/BM and the activity of intracellular signaling proteins from *in vivo* VO studies. The output of the Bayesian calibration is a PDF for the baseline weight of each input reaction.

We validated the calibration using independent studies of the infusion of individual agonists that can stimulate hypertrophy as well as combinations of those agonists with pharmacologic blockers of various signaling pathways. For each validation simulation, we performed a Monte Carlo simulation (MC), with one thousand iterations (N=1000). On each iteration, we selected a set of baseline reaction weights $\left( w_{AngII}^0, w_{NE}^0, w_{ET1}^0, w_{background}^0 \right)$ from their calibrated PDFs. We simulated the infusion of agonists by increasing the weight of the corresponding reaction while keeping the rest of the reactions at their selected baselines. When simulating the infusion of agonists at saturation doses, we increased the weight of the corresponding reaction to its maximum value of one. Otherwise, we randomly selected a fold-change value from a normal distribution matching the experimentally reported mean and deviation of the agonist concentration in plasma and increased the input reaction weight by that same fold changes. To simulate the effect of pharmacologic blockers, we set the maximum activity of the target receptor to zero [85]. For the simulation of VO, we randomly picked time-varying curves of AngII, NE, ET1, ANP, BNP, and $\varepsilon_f$ from their respective PDFs on each iteration. Although some of the agonist and blockers had significant effects on heart rate (HR) and blood pressure (BP) in the corresponding studies, we did not model those effects here.

In prior studies using large-scale signaling networks, our group compared model-predicted changes (increase, decrease, or no change) for individual outputs or signaling intermediates to published measurements (significant increase, significant decrease, or no significant change) not used for model construction and calibration [85]. Here, we adapted that approach slightly to account for the probabilistic nature of the Bayesian calibration. Each model simulation now produces a distribution of possible outcomes; if at least 75% of those simulations predicted an increase, we treated that as a model-predicted increase. Predicted decreases were handled similarly. We considered the model prediction to be validated if most available studies on the simulated intervention reported a significant increase or decrease in the same

output. As a more quantitative metric, we also counted how many published studies reported a mean change that fell within the 50% and 90% confidence intervals (CI50 and CI90) of the model prediction.

**2.8.1. *Simulation of infusion-induced hypertrophy.*** We performed *in silico* simulations of infusion experiments of ISO and ANGII in rats, and NE in dogs. We quantified the hypertrophic effect of infusion as the relative growth with respect to baseline:

$$Normalized\ Growth = \begin{cases} \frac{LVM/BM^{final} - LVM/BM^{initial}}{LVM/BM^{initial}} & (experiment) \\ \frac{CellArea^{final} - CellArea^{initial}}{CellArea^{initial}} & (model) \end{cases}$$

(6)

We collected fifteen experimental datasets (mean±SD) from twelve independent studies of ISO infusion in rats at rates >1mg/kg/day, a rapid-growth-inducing saturation dose [86–97]. We simulated these experiments by imposing the maximum input weight of 1 on the ISO node to simulate its infusion at saturating rates and comparing the steady-state growth prediction to experimental data collected at termination times longer than one week, at which maximum growth was reached in all reviewed experiments.

AngII infusion in rats at rates of 100–200 ng/kg/day is a standard rodent model of hypertension. At this infusion rate, the plasma concentration of AngII increases two- to five-fold [98–104]. We collected 20 experimental datasets from 14 independent studies in rats, measured at 1 week, 2 weeks, or 4 weeks of infusion [98–111]. We simulated these experimental conditions by increasing the AngII stimulation by 3.5 ± 1.5 fold.

We found four independent reports of experimental NE infusion in dogs, each with a different dosage and termination time [112–115]. Each of the experiments in dogs reported the plasma concentrations achieved, and we therefore simulated each individual experiment by imposing the same fold increase on the NE network input as the reported fold change in plasma NE, for the same duration as the experiment.

**2.8.2. Replication of drug treatment of infusion-induced hypertrophy.** To validate the capacity of our model to reproduce cross-talk between different signaling pathways, we also simulated experiments where agonist infusions were paired with β-adrenergic receptor blockers (βB), angiotensin receptor blockers (ARB), or endothelin receptor antagonists (ERA). In each case, we increased the input weight of the infused hormone while keeping the rest of the inputs at their baseline and setting the activity of the targeted receptor at zero. We quantified the drug effect on infusion-induced hypertrophy as the chronic difference between the growth during infusion plus blocker and growth during infusion alone measured at the same endpoint, normalized to the initial dimension:

$$Normalized\ Drug\ Effect = \begin{cases} \frac{\frac{LVM}{BM}^{final}_{infusion+drug} - \frac{LVM}{BM}^{final}_{infusion}}{LVM/BM^{initial}} & (experiment) \\ \frac{CellArea^{final}_{infusion+drug} - CellArea^{initial}_{infusion}}{CellArea^{initial}} & (model) \end{cases}$$

(7)

We collected eight datasets from four independent articles studying the effect of βB on ISO-induced hypertrophy [88,95,116,117], five experimental datasets from five independent publications studying the effect of ARB on ISO-induced hypertrophy [86,96,97,117,118], three experimental datasets from three experimental studies on the effect of ERA on AngII-induced hypertrophy [111,119,120], and one experimental point from a publication studying the effect of ERA on NE-induced hypertrophy [121]. All of these experiments were performed in rat models.

**2.8.3. Validation of additional model outputs.** Cardiac hypertrophy is typically associated with changes in expression of ANP, BNP, αMHC, βMHC, and SERCA that are collectively termed the fetal gene program [19,122,123]. The network model employed here predicts each of these outputs, but no data on their changes were included in the Bayesian calibration process. As an additional independent validation of the performance of the calibrated network model, we therefore compared model-predicted changes with experimental reports of significant changes in ANP, BNP, αMHC,

βMHC, and SERCA protein abundance in tissue extracted following infusion experiments. Because the model predicts gene expression while the experiments measured protein abundance, we did attempt quantitative comparisons for these outputs but rather simply assessed whether they increased significantly, decreased significantly, or did not change significantly.

### 2.9. Administration of receptor blockers during VO

Several drugs currently employed in patients with heart failure have been tested in animal models of experimental VO. While we expect that some of those drugs have important secondary effects not represented in the current model, we simulated available experiments to further explore its capabilities and limitations. Model predictions were compared to data from treatment of experimental VO with βB, ARB and ERA. We collected six experimental data points on the early ($t/\tau \leq 1$) and two on the chronic ($t/\tau > 1$) effects of βB in VO from three articles on experimental MR in dogs [15,16,80] and two articles on experimental VO in rats [14,124]. We collected five experimental datasets on the effect of ARB on VO-induced hypertrophy from three VO experiments in rats [42,46,50], and one from MR experiments in dogs [29]. The effect of ERA on VO was compared to four experimental datasets from four independent articles on experimental VO in rats [125–128].

## 3. Results

### 3.1. Cross-species data integration and probability distribution of hormonal stimuli

The best-fit characteristic time constant for mass growth was ($\tau_M$) 1095 hours for dogs and 283 hours for rats. The ratio of the species time constant was $\frac{\tau_{M_{dog}}}{\tau_{M_{rat}}} = 3.86$. When scaled by this ratio, data from rats and dogs were reasonably congruent; variability within individual studies and/or between different studies in the same animal model were generally larger than the differences between means measured from the two different species at similar relative times (Fig 1). Interestingly, the ratio of growth time constants computed here is close to the ratio of the reported heart rates (HR), which is often used for allometric scaling [129]. With an average HR of $363 \pm 14$ *bpm* for rats and $95 \pm 4$ *bpm* for dogs, the HR ratio was $\frac{HR_{rat}}{HR_{dog}} = 3.82 \pm 0.22$.

### 3.2. Probability distributions of fiber strain during VO

We combined available data on normal heart mechanics and volume-overload hypertrophy in dogs with a simple geometric model to estimate the probability distribution of end-diastolic strain ($\varepsilon_f$) relative to an unloaded state over the course of experimental VO. The PDF displays the expected trends over time: an acute increase in strain owing to the sudden increase of $V_{ED}$ when mitral regurgitation is first introduced, followed by a gradual decrease driven by the compensatory hypertrophic response (Fig 3). As $V_{ED}$ and LVM curves reach a plateau in the chronic stages of VO, $\varepsilon_f$ also stabilizes. Models in which mechanical strain is the sole driver of cardiomyocyte growth only produce a stable hypertrophied state if strain returns to its baseline level, or the homeostatic strain level is allowed to adapt [13,130]. Interestingly, our data analysis suggests that in 75% of the cases, strain falls below its original baseline level in chronic stages of experimental VO (Fig 3c), with 50% of the cases passing below this threshold relatively early (t/$\tau \leq 1$ corresponding to the first 12 days in rats and 6 weeks in dogs). The final chronic level of strain showed a strong inverse correlation to chronic LVM fold change (PCC=-0.78), suggesting that myofiber stretch is more likely to fall below baseline in cases with the greatest mass increase (Fig 3d).

### 3.3. Calibration of relative weights for input stimuli to the hypertrophy network model

In this work we allowed the key network inputs to have different weights, and calibrated those weights using experimental data. Preliminary screenings of the parameter space revealed that the baseline weight of the myoStrain input $\left(w^0_{myoStrain}\right)$

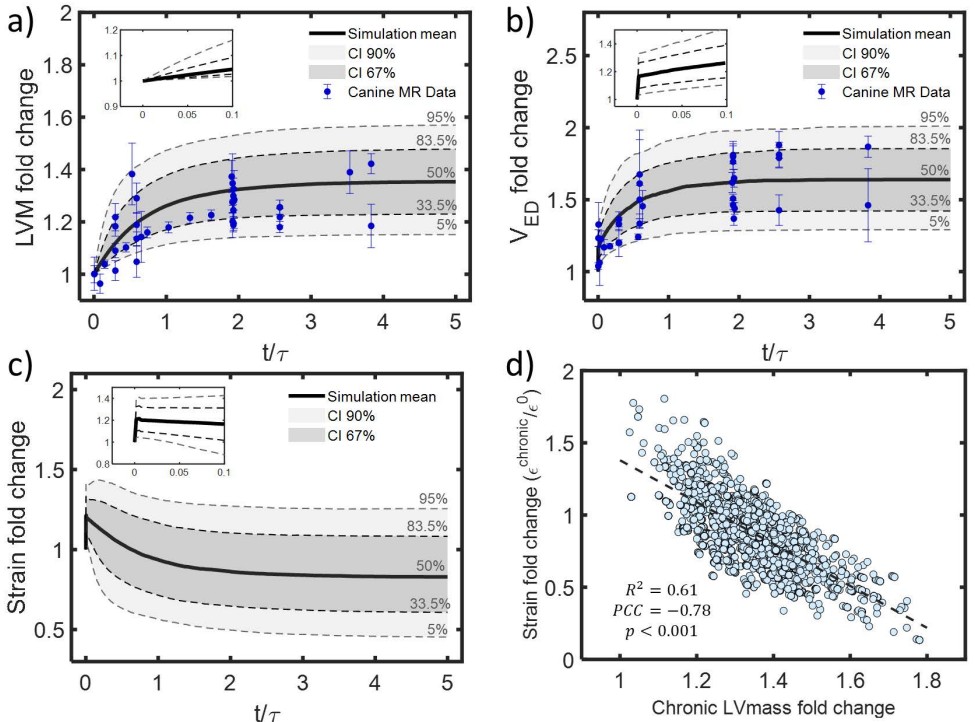

**Fig 3. Changes in volume, mass, and end-diastolic strain during experimental MR in dogs.** a) Fitted data on fold changes show a) a 36% increase in LVM [15,16,20–33] and b) 67% increase in LV end-diastolic volume ($V_{ED}$) [16,17,20–26,29,32,33,61,78,80,82,83,131–136] on average at steady-state. c) In a spherical model, these changes imply a 20% increase followed by a gradual decrease below baseline in end-diastolic strain on average, with a wide range of trajectories possible for different combinations of mass and volume change. d) Calculated chronic strain levels are lower when LVM increases more. Insets in panels (a-c) show acute changes plotted on a magnified time axis.

strongly influenced the probability of producing unlikely predictions (i.e., those that contradict experimental observations) such as spontaneous growth reversal despite continued overload and runaway growth that never stabilized. We therefore ran a series of MCMCs at fixed values of $w^0_{myoStrain}$ while randomly sampling the rest of the input reaction weights as described in section 2.6.1. We found that constraining the baseline weight of the myoStrain input within the range $0.05 \leq w^0_{myoStrain} \leq 0.06$ minimized the number of unlikely solutions. For $w^0_{myoStrain} > 0.06$, the solutions tend to be dominated by the myoStrain input, resulting in a reversal of growth at later time points despite continuing simulated overload, while the solutions for $w^0_{myoStrain} < 0.05$ were dominated by adrenergic stimulation, increasing the chances of runaway growth (Fig H in S1 Text). The results presented here were therefore obtained with a fixed $w^0_{myoStrain} = 0.06$, which was associated with the highest mean likelihood among the myoStrain values we tested.

The posterior PDF of the remaining input reaction weights converged to $w_{ANGII} = 0.010 \pm 0.002$, $w_{NE} = 0.033 \pm 0.014$, $w_{ET1} = 0.056 \pm 0.025$, and $w_{background} = 0.031 \pm 0.022$. We calculated correlations among these input weights across the 20,000 simulations from the final round of MCMC simulations. The strongest correlation was between $w^0_{ET1}$ and $w^0_{background}$ with a Pearson PCC=-0.42, while all other PCC magnitudes were below 0.3 (Fig I in S1 Text). The PDF of the strain-to-Myostrain mapping parameter $C_{myo}$ converged to $C_{myo}$=5.86 ± 1.38. Among all nodes in the network, the mean predicted chronic activation level at t/τ = 3 (Fig 4) increased most for Ca++/calmodulin-dependent kinase (CaMK) and calcineurin (CaN) and decreased most for integrin and focal-adhesion kinase (FAK).

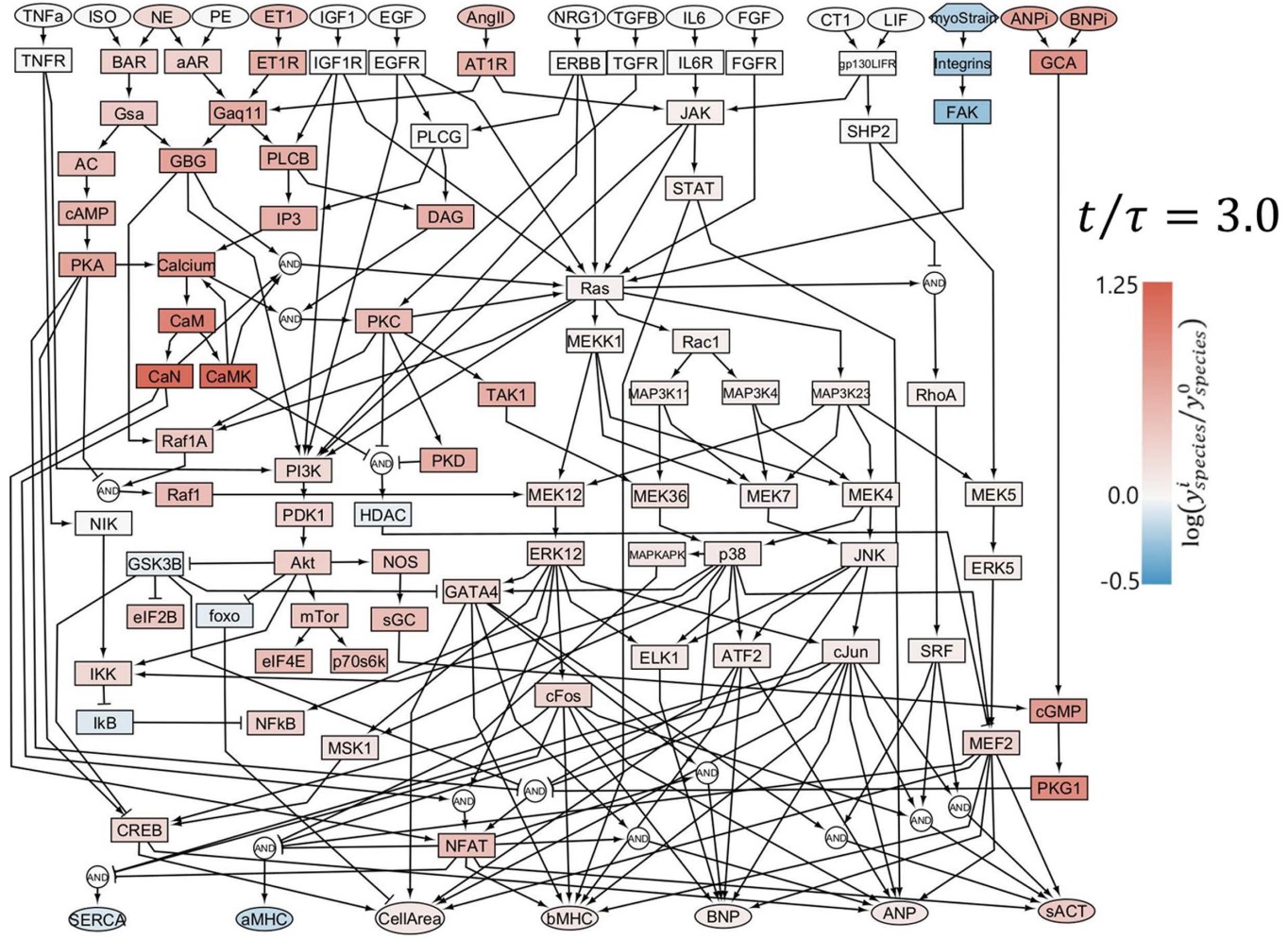

**Fig 4. Predicted chronic activation of the cardiomyocyte signaling network during VO.** Colors indicate the logarithm of the fold change in activity of each node in chronic stages of VO relative to baseline, with red indicating an increase and blue indicating a decrease.

## 3.4. Validation

First, we generated a forward MC simulation of untreated VO as described in section 2.7. Briefly, we performed N=1000 simulations of growth until steady-state (t/τ = 5) in untreated VO. For each iteration we selected a set of baseline reaction weights from their calibrated (posterior) PDFs, and a set of time-varying curves for neurohormonal stimulation and stretch from their respective PDFs [134]. The model predicted a wide range of possible growth curves for untreated VO (Fig 5a). On average, normalized growth in CellArea (Eq 6) was 47%, with 50% of simulations (CI50) predicting growth between 34% and 60%, and 90% of simulations (CI90) predicting growth between 17% to 83%. More than 75% of our simulations predicted chronic increases in ANP, BNP, and βMHC and decreases of SERCA and αMHC; these predictions agreed well with experimental measurements of protein abundance from cardiac tissue extractions in experimental VO (Fig 5b).

 **3.4.1. Simulation of infusion-induced hypertrophy.** We validated the calibrated model against data from independent experiments not used in its calibration that infused hypertrophic agonists over periods of weeks and measured the resulting hypertrophy. Our model predicts that infusion of ISO at saturating doses produces a mean LV growth of 40%, with 50% of simulations (CI50) falling between 33 and 50% growth, and 90% of simulations (CI90) falling

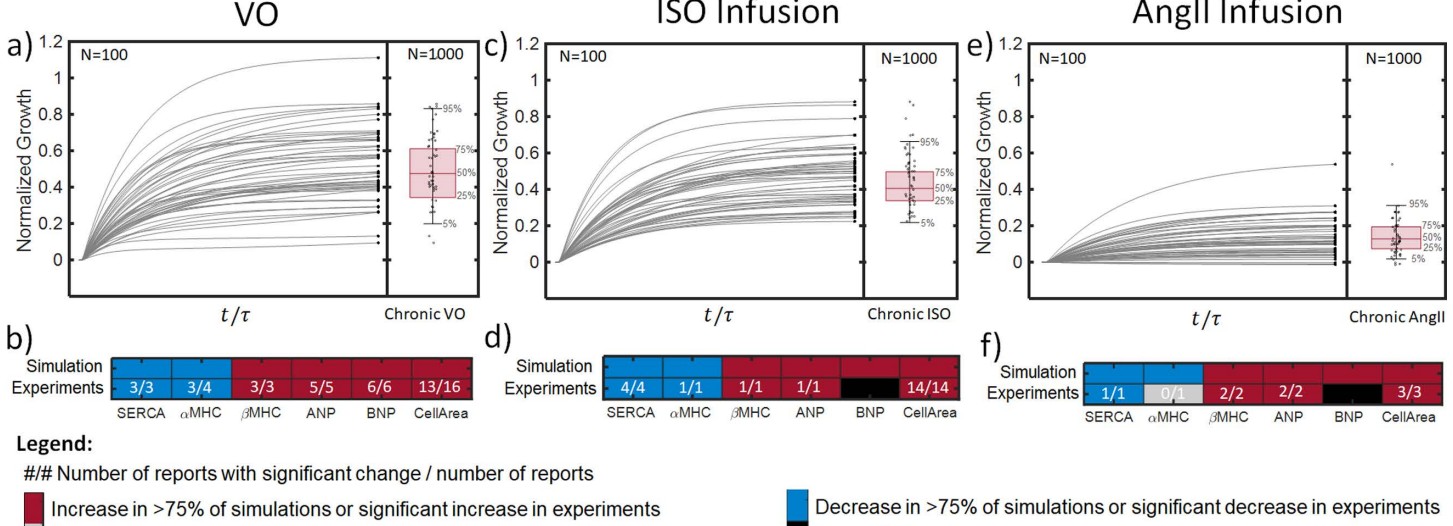

**Fig 5. Results of MC simulations of VO and agonist infusion.** Top panels show normalized growth in CellArea for 100 representative simulations (gray lines), with terminal values indicated by black circles; adjacent boxplots reflect the mean, CI50, and CI90 of terminal values for 1000 simulations of a) Untreated VO, c) ISO infusion, and e) AngII infusion. Bottom panels contain heatmaps comparing the predictions of changes in signaling network outputs (SERCA, αMHC, βMHC, ANP, BNP, and CellArea) to experimental data following b) Untreated VO, d) ISO infusion, and f) AngII Infusion. Red indicates that more than 75% of simulations predicted an increase in the output, or that the majority of studies reported a significant increase. Blue indicates a decrease in >75% of simulations or the majority of experiments. Gray indicates the absence of consistent changes in simulations or significant changes in experiments. White numbers indicate the number of studies reporting a significant change and the total number of studies reporting data for each output.

in the range 23–68% (Fig 5c). These predictions show good agreement with experimental data, with all available studies reporting a significant hypertrophic effect of ISO at infusion rates >1 mg/kg/day (Fig 5d). Four out of 15 studies reported mean increases in LVM/BM that fell within our CI50, and 12 of 15 reported means fell within the predicted CI90 (Fig 6a) [86–97]. Our model also predicted increases in ANP, BNP, and βMHC and decreases in SERCA and αMHC in more than 75% of simulations of ISO infusion (Fig 5d), which is consistent with the significant changes reported in all experimental measurements of ANP, βMHC, SERCA, and αMHC protein abundance during ISO infusion (Fig 5d) [88,93,95].

Our model predicted that increasing the serum concentration of AngII by *3.5 ± 1.5*-fold would produce long-term growth of 13% on average, with 50% of simulations (CI50) in the range [7%,19%], and CI90 of [1.8%,31%] (Fig 5e). In more than 75% of simulations, AngII activated the fetal gene program, decreasing SERCA and αMHC while increasing βMHC, ANP, and BNP (Fig 5f). This prediction agreed with all experimental data collected for validation, except for a single study that found no significant difference in βMHC protein abundance relative to control [105,106,109]. When comparing to data collected at different time points during the course of AngII-induced hypertrophy, more than 75% of simulations predicted growth at every time point, and the magnitude of predicted growth increased gradually with time (Fig 6b). These predictions compared well to independent experimental data: at the earliest time point, 7 of 9 experimental means fell within the predicted CI90; while at later time points all experimental means fell within the CI90. The simulations appeared to under-predict growth somewhat, with no experimental means falling inside the CI50 at 1 week (t/τ = 0.6), 4 out of 8 datasets within the CI50 at 2 weeks (t/τ = 1.2), and 1 out of 3 experimental points collected at one month of AngII infusion (t/τ = 2.4) falling within the predicted CI50 [98–111]. Variability in the experimental data was large compared to the effect size of AngII treatment, and the CI90 of the model predictions spanned a similar range to the experimental data, illustrating the potential advantages of the Bayesian approach.

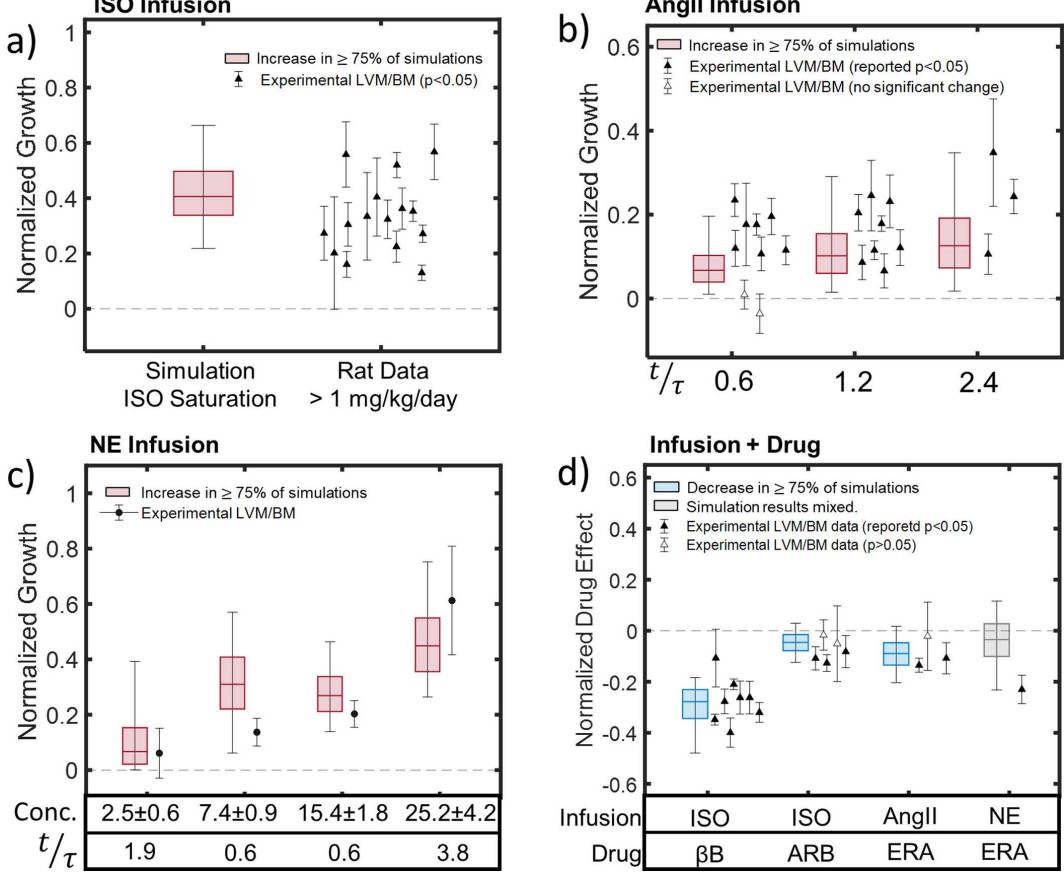

**Fig 6. Validation of model predictions of growth during infusion of hypertrophic agonists with and without receptor blockers.** a) Boxplot shows the mean, 50% confidence interval, and 90% confidence interval for predicted normalized growth in 1000 MC simulations of ISO infusion with the calibrated network. Twelve of 15 experimental means (plotted as individual points with error bars) from [86–97] fell within the CI90. b) Model-predicted CI90 for growth during AngII infusion encompassed most published values at the earliest time point and all reported experimental means at later time points [98–111]. c) Simulations of four NE infusion experiments in dogs with different doses, fold increases in plasma concentrations (Conc.), and termination times ($t/\tau$) showed reasonable agreement with the limited available data [112–115]. d) Simulations of agonist infusion plus receptor blockers replicated reported significant effects of β blockers (βB) on ISO-induced hypertrophy infusion [88,95,116,117], and smaller effects that were significant in some but not all studies for ARB administration during ISO infusion [86,96,97,117,118] or ERA administration during AngII infusion [111,119,120]. By contrast, the model prediction of nearly equal likelihood that ERA increases or decreases NE-induced growth appears to contradict a significant reported decrease in the only available experimental study [121]. Red coloring of boxplots indicates increases in >75% of simulations, blue indicates decreases in more than 75%, and gray indicates mixed results. Points with error bars indicate experimental mean±SD for studies reporting statistically significant effects (filled markers) or non-significant effects (open markers), from studies in dogs (circles) or rats (triangles).

Simulations of NE infusion showed reasonable agreement with experimental data, displaying growth in >75% of simulations at all plasma concentrations and increasing growth with higher concentration or longer durations (Fig 6c). The experimental means for all simulated infusion protocols fell within the CI90 of the model predictions, and 2 out of 4 fell within our CI50. We note that the lowest plasma concentration simulated here fell within the range of NE levels observed during experimental VO, while the other simulations employed concentrations well above the calibrated range.

**3.4.2. Simulations of agonist infusion plus receptor blockade.** In addition to testing model predictions of the effects of infusing individual hypertrophic agonists, we validated the model against experimental data on combinations of those agonists with various receptor blockers. Our model predicted that βB administration would reduce ISO-induced hypertrophy by 27%, with 50% of simulations falling in the range [-36%, -22%] and 90% of simulations falling in the range

[-47%, -18%] (Fig 6d). These predictions showed good agreement with experiments, with 7 out of 8 reported means falling within our CI90, 5 falling within the CI50, and all experiments reporting a statistically significant effect of βB treatment [88,95,116,117].

The model predicted a much smaller effect of ARB administration on ISO-induced growth, with a mean reduction in CellArea of 4.5%, CI50 of [-8%, -1.5%], and CI90 of [-12%, +3%] (Fig 6d). Consistent with the fact that our CI90 included more predicted decreases but some predicted increases, only 3 out of 5 studies reported a significant effect of ARB administration on hypertrophy. All reported means fell within the model-predicted CI90, and 3 out of 5 within the CI50 (Fig 6d) [86,96,97,117,118]. Similarly, our model predicted that ERA would reduce AngII-induced hypertrophy by an average of 9%, with a CI50 of [-13%, -4.7%] and a CI90 of [-20%, +2%] (Fig 6d), which was consistent with significant reported reductions in 2 of 3 studies [111,119,120]. 2 out of 3 experimental means fell within the model-predicted CI50, and all reported means fell within the model CI90.

When simulating the combination of NE infusion and ERA administration, our model predicted a negligible impact on hypertrophy, with a 3% reduction on average with respect to untreated NE-infusion, a CI50 of [-10%, +3%] and a CI90 of [-23%, +12%]. This contrasts with the sole experimental report of a 23% decrease, although the mean of that study did still fall within our CI90 [121].

### 3.5. Administration of receptor blockers during VO

We explored the capacity of the model to predict the effect of β blockers (βB), angiotensin receptor blockers (ARB), and endothelin receptor antagonists (ERA) in the context of experimental VO. In these simulations we ignored any possible secondary effects of the blockers on circulating hormone levels or myocardial strain. Experiments find no effect of β-blockers on ventricular mass in early stages of VO ($t/\tau < 0.7$) in dogs or rats [14,15,80,124], but chronic use of this drug ($t/\tau = 2.7$) appears to exacerbate hypertrophy in both animal models [14,16,124]. In our simulations, β-blockers produced increases and decreases in mass with equal frequency, resulting in a negligible mean effect on average at both early and late stages (Fig 7a). Although

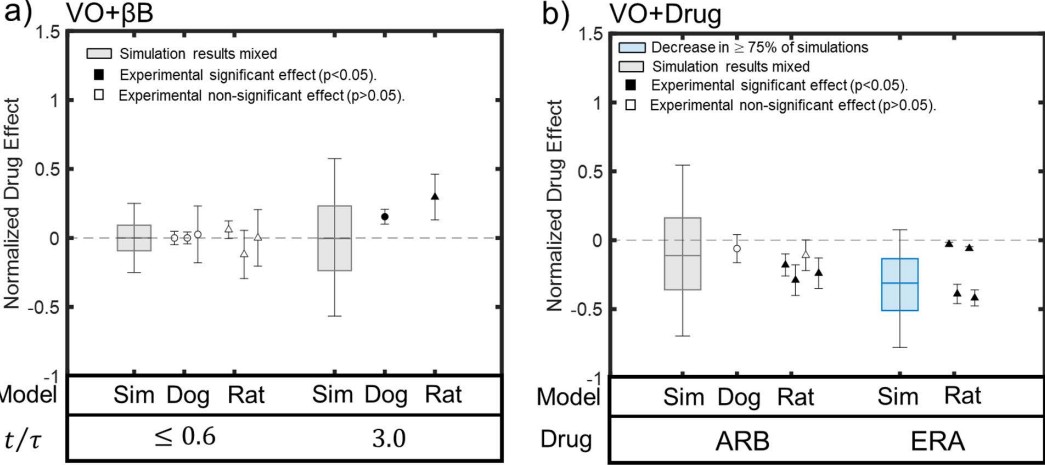

**Fig 7. Effects of receptor blockers on VO-induced hypertrophy in published experiments and the calibrated model.** a) Simulations of β blockers (βB) at both early ($\frac{t}{\tau} \leq 0.6$) and late stages ($\frac{t}{\tau} = 3$) of VO, simulations (boxplots) were equally likely to predict increases or decreases, with no net effect on average. MR experiments in dogs (circles) [15,16,80] and VO in rats (triangles) [14,124] also found no significant effect early but a further increase in growth at later timepoints relative to untreated VO. b) Simulated administration of angiotensin receptor blockers (ARB) also produced mixed results, consistent with mixed results in published studies [29,42,46,50]. By contrast, simulated ERA administration decreased VO-induced hypertrophy in the majority of simulations, and in all four available experiments [125–128]. Blue coloring of boxplot indicates predicted decrease in >75% of simulations, gray boxplots indicate mixed simulation results, filled markers indicate experimental means associated with statistically significant effects and open markers indicate means associated with non-significant effects.

our simulations would not have anticipated the significant reported increase in mass in the two studies of chronic administration, experimental means from all studies and time points did fall within the model-predicted CI90.

When simulating the effect of ARB administration on VO-induced hypertrophy, our model produced mixed results, with only 62% of the simulations predicting a reduction in hypertrophy and a mean normalized drug effect of -11%. These mixed predictions of ARB effects are consistent with the fact that two VO experiments (one on rats and one on dogs [29,50]) reported no significant effect of ARB on hypertrophy while three other rat experiments reported a significant decrease [42,46,50]. Experimental means from all of these studies fell within our predicted CI50 [-36%,+16%].

Our model predicted that ERA dampens VO-induced hypertrophy in 90% of the cases, with a mean effect of -31%, CI50 of [-51%,-13%] and CI90 of [-77%,8%]. These predictions agree with the significant hypertrophy-inhibiting effects of ERA reported in all rat studies reviewed here [125–128]. Experimental means from all 4 studies fell within our predicted CI90, and half within our CI50.

### 3.6. Sensitivity analysis

The sensitivity analysis revealed that at late simulated time points, all network outputs were far more sensitive to ET1 than to any of the other network inputs examined in this study (Fig 8). This suggests that, in chronic stages of VO, variation in the circulating level of ET1 within its reported range has the largest influence on cardiomyocyte size and gene expression of any individual hormonal or mechanical growth factor. According to this analysis, higher levels of ET1 in late stages

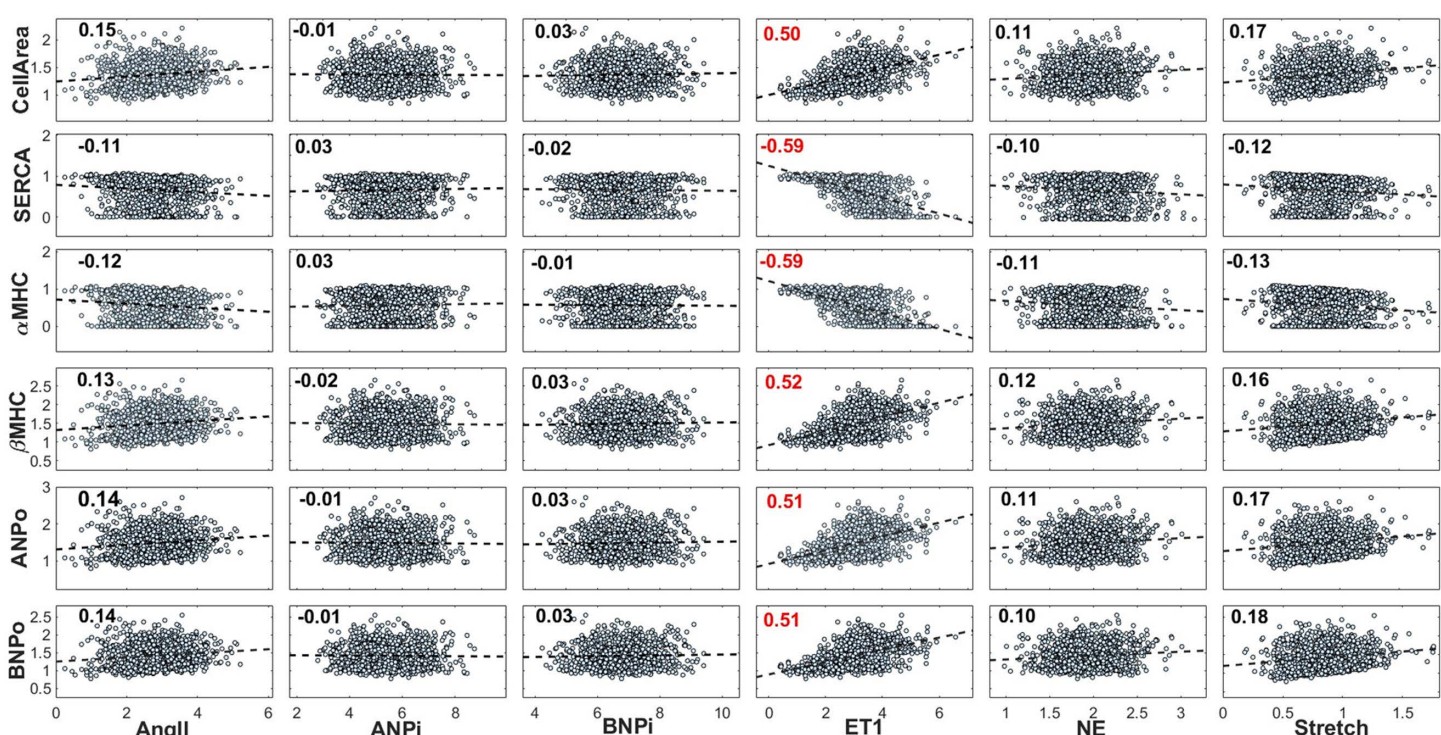

**Fig 8. Sensitivity analysis of network outputs to calibrated network inputs in chronic stages of VO.** Each marker represents the final state of one of 1,000 Monte Carlo simulations of VO. A linear regression model (dashed line) is fitted to each output-input pair, and the Pearson PCC is displayed on the upper left corner of each. Among the inputs known to vary substantially over the course of VO, ET1 displayed the largest influence on all outputs in the late stages of overload.

of VO are associated with greater cardiomyocyte growth, greater myocardial production of βMHC, ANP and BNP, and reduced myocyte synthesis of SERCA and αMHC.

## 4. Discussion

This study used a Bayesian approach to integrate a large body of experimental VO data in rats and dogs and calibrate a well-established model of hypertrophic signaling in the heart. The resulting model captures both the temporal variations in neurohormonal and mechanical stimuli during VO and the relative responsiveness of cardiomyocyte signaling pathways to those stimuli. As a result, the model not only reproduced the data on hypertrophy used to calibrate it, but also correctly predicted responses not used in calibration, such as the re-expression of the fetal gene program during VO. Furthermore, following calibration against data from 70 studies of VO, the model performed remarkably well when validated against data from independent studies of other interventions that trigger some of the same signaling pathways, such as infusion of hypertrophic agonists with and without administration of receptor blockers. More specifically, 50 of 56 (89%) reported mean changes in LVM/BW collected from those studies at various doses and times fell within the 90% confidence interval predicted by the model for its CellArea output.

Our primary motivation in developing a systems-level model of hypertrophy during VO is that clinical conditions such as mitral regurgitation trigger changes in multiple hypertrophic stimuli, including myocyte stretch/stress as well as circulating levels of hormones known to modulate hypertrophy. Furthermore, most patients receive multiple therapies – such as administration of β-blockers and mitral valve repair or replacement – that themselves modify different hypertrophic stimuli. Thus, predicting the effects of any potential treatment or combination of treatments requires a model that can anticipate the crosstalk amongst relevant signaling pathways. In this respect, the validations shown here against experiments where agonist infusion was combined with a receptor blocker (Fig 6d) are particularly promising. In most cases, both the mean response and the range of expected responses predicted by the model agreed with available experimental data; the one exception was the combination of NE + ERAs, where our model prediction disagreed with the lone study available for comparison.

We also explored model predictions of drug treatment during experimental VO, but did not treat those simulations as formal validations because a multiscale model would be needed to capture the expected secondary effects of the drugs on hemodynamics and ventricular mechanics. While simulations of ARB and ERA treatment during VO matched data fairly well, the limitations of the model were more apparent when simulating β-blockade (Fig 7). Chronic β-blockade during experimental VO appears to exacerbate hypertrophy after several months in dogs and rats [16,21,33,124]. By contrast, in our calibrated model NE had limited influence on hypertrophic signaling during VO, so simulated β-blockade had little effect on hypertrophy. The exaggerated NE concentration levels required by Laks et al. (1973), King et al. (1987) and Stewart et al. (1992) to produce LVM increments above 20% support the idea that serum levels measured during VO should exert limited influence on hypertrophy [112–114], and β-blockade does have minimal impact in isolated cardiomyocytes and short-term VO experiments [14,15,80]. We therefore hypothesize that worsening of hypertrophy by long-term β-blockade arises indirectly through their depression of LV contractility, an effect that was not included in the model employed here. Evidence supporting that hypothesis includes the fact that β-blockade partially restores FAK phosphorylation levels during experimental VO, suggesting increased activation of stretch-modulated hypertrophic pathways [14–16,60]. A further complication when modeling β-blockers is that they appear to improve LV function, symptoms, and survival in patients with chronic MR and heart failure [137,138] due to remodeling of the signaling pathways themselves, another effect not modeled in the present study.

The Bayesian calibration approach proposed here results in a model that predicts a distribution of expected responses for any simulated intervention. When using a model to pre-screen potential therapies, this approach could have the important advantage of providing information on both the expected effect size and the expected variability in the response. This information could be used to prioritize for further development potential treatments that are predicted to

produce the desired effect in the vast majority of cases, something not possible with traditional models that predict only a mean response. Alternatively, if measurable pre-treatment factors distinguish a subpopulation of simulations with large predicted responses for an otherwise unimpressive potential therapy, this information might motivate further exploration of its potential in a specific subgroup of patients. In the model presented here, the variability in any predicted response reflects both the known topology of the signaling network being modeled and the variability in the data used for calibration. In most cases where sufficient comparison data were available to make an assessment, the spread of data from independent studies used for validation was very similar to the range of responses predicted by the model; this suggests that the calibration process worked as intended. Viewed through this lens, even the one clear validation failure (Fig 7, NE+ERA) might not be so bad. The model predicted a very small effect on average but high variability in that response; although the one published study reported a significant reduction of NE-induced hypertrophy by ARBs, its mean fell well within the model-predicted range. If the experiment were repeated several more times, would this finding persist? Or would other experiments produce a mix of significant and non-significant findings, as occurred for some of the other combinations modeled?

The calibrated and validated model suggests some insights regarding hypertrophic signaling in chronic VO that are not apparent from the structure of the signaling network alone. For instance, although any of the agonists discussed here are capable of stimulating cardiac hypertrophy, our sensitivity analysis showed ET1 to be the primary driver of hypertrophy during chronic stages of VO (Fig 8). Consequently, the model predicted that ERA treatment reduced VO hypertrophy in the majority of simulations, while other blockers produced mixed results (Fig 7). These predictions not only agree with the experimental data we collected for validation [29,42,46,50,125–128], but also with the observations by Leskinen et al. 1997 and Fareh et al. 1996 on combined treatment of VO in rats with ARB and ERA, that suggest ET1 stimulation is more important in regulating the long-term myocyte adaptative response to VO than AngII or stretch [139,140]. Despite the apparent benefits of ERAs in experimental VO and their proven vasodilator effect, their clinical use has been limited by potentially severe side effects, such as alterations of liver function, anemia, and edema. The most common clinical application of ERAs is to treat pulmonary artery hypertension, and recent efforts point to the development of selective ERAs for treating persistent hypertension [141].

One of the most interesting implications of our analysis is our prediction that in most cases of VO, the combination of early overstretch and sustained neurohormonal activation trigger sufficient hypertrophy to drive stretch below its baseline levels. This prediction did not derive from the structure of the signaling network but rather from the Bayesian approach to integrating published data on observed LV mass and end-diastolic volume increases across a large number of experimental studies (Figs 2 and 3). This prediction may explain the otherwise puzzling depression in the activity of mechano-transduction pathways in the context of volume overload. While FAK phosphorylation is elevated in pressure overload and aortic valve regurgitation relative to baseline, FAK phosphorylation is reduced in VO despite elevated LV volumes that are commonly assumed to indicate elevated levels of myocyte stretch [15,16]. In previous work, we identified Ras as a relevant hub responsible for the crosstalk of multiple pathways [19,73]. In the VO simulations presented here, Ras was a critical node integrating the competing effects of mechanical stretch and neurohormonal inputs. During early VO, elevated stretch and neurohormonal stimulation combined to drive strong activation of Ras, while in chronic VO reduced stretch and FAK activity counteracted continuing neurohormonal stimulation, producing a low level of Ras activation (Fig 4 and S2 Material).

## 5. Limitations and future directions

The ability to predict not only mean responses but also the uncertainty around those predictions is a major advantage of Bayesian approaches. However, the approach described here may overestimate variability of the predicted responses, for several reasons. First, we treated the levels of the various circulating hormones as independent of each other; for example, a given VO simulation might randomly combine input curves specifying very large changes in one input and

very small changes in the others. However, in reality most reports suggest that the expression of circulating neurohormones is correlated to the severity of cardiac insult, and therefore to each other. The current model also lacks output-to-input feedback loops. For example, the strain-time curve is imposed through random sampling, with no previous knowledge of the growth response, and the likelihood of a given CellArea prediction is evaluated against all available data on mass increases, not just those occurring at a similar level of strain. To address these sources of variability, we could introduce additional experimental data and covariance relations to the likelihood evaluation. Another source of variability that would be more difficult to address is the fact that the data used to calibrate the model were gathered from studies in different animal models, performed by different groups. These studies typically began with healthy animals and introduced a severe or moderately severe overload at a known point in time. However, factors such as variations in overload severity, species differences, and sex differences likely contributed to the variability in predictions in the final, calibrated model.

Another limitation worth mentioning is that the simulations of both agonist infusions and receptor blockers ignored secondary effects on hemodynamics, LV mechanics, and strain. Known experimental responses that were neglected due to this simplification include an increase in heart rate during ISO and NE infusion, an increase in systemic blood pressure with ISO infusion in some studies [92,96,116] and with high doses of NE [114], vasoconstriction by AngII, vasodilation by ARB and pulmonary vasodilation by ERA, and effects of adrenergic agonists and receptor blockers on LV contractility and mechanics. One way to address these limitations would be to incorporate the calibrated myocyte signaling model developed here into a multiscale model of cardiovascular function. A multiscale model could use tissue and organ-scale ventricular models to update the estimations of myocardial strain as growth proceeds, and represent known feedback between systemic hemodynamics, neurohormonal alterations, drug effects, and heart loading.

We assumed that changes in neurohormonal circulating concentrations are proportionally transduced into receptor activity, which is likely true only when receptor availability is high and binding is non-competitive, neglects changes in receptor abundance, and ignores possible differences in hormonal concentrations between the bloodstream and the immediate cellular environment. Hormone concentrations in myocardial tissue extractions can be several orders of magnitude larger than circulating concentrations. However, they show similar trends, suggesting that the circulating and local values are at least correlated [22,30]. Ultimately, calibrating the model to widely accessible data such as serum concentrations improves its translational value.

## 6. Conclusions

We employed a Bayesian approach to combine data from 70 studies on experimental volume overload in dogs and rats and use it to calibrate a network model of hypertrophic signaling in cardiomyocytes containing nearly 200 reactions. The calibrated model reproduced many key results from 43 independent studies not used in its calibration, including infusion of hypertrophic agonists alone or in combination with receptor blockers and administration of multiple heart failure drugs in the setting of experimental VO. In nearly all cases, if the majority of model runs for a given condition predicted a consistent effect, the majority of experimental studies found the same effect; where enough studies of a given intervention were available for comparison, the variability among those studies was also similar to the range of responses predicted by the model. These results suggest that the approach presented here could be useful in future simulations of novel potential treatments for VO. The model suggests that changes in cell size and activation of the fetal gene program in the chronic stage of VO are particularly sensitive to Endothelin1 receptor activity. The calibrated model also suggests that growth in experimental VO is mostly driven by the neurohormonal response, with eccentric hypertrophy reducing initially elevated myocardial strain values below baseline fairly quickly in most cases. This prediction provides a plausible explanation for the depression of mechano-transduction signaling pathways in experimental VO, despite the widespread conception of volume overload hypertrophy as driven by myocyte overstretch.

## Supporting information

**S1 Text. This Word document provides additional information on the methods employed in the study.** This document includes tables of the literature sources that provided data for each model input and output used in calibration and validation, more details on the fitting process used to generate probability density functions for inputs to the myocyte signaling network, and information on where covariance was present among the model inputs and how it was handled. (PDF)

**S1 Material. This Excel workbook provides the tables needed to re-create the signaling network model used here with the freely available Netflux software ([https://github.com/saucermanlab/Netflux](https://github.com/saucermanlab/Netflux)).** The tables specify connectivity of the nodes in the network as well as the numerical parameters governing each reaction in the network. (XLSX)

**S2 Material. This GIF contains a color-coded animation of the states of every node in the calibrated myocyte signaling network as they evolve over the course of simulated volume overload (VO).** Red colors indicate a higher level of activity during VO relative to baseline, while blue colors indicate a lower level of activity during VO relative to baseline. (GIF)

## Acknowledgments

The authors thank Dr. Pim Oomen, University of California Irvine, for his support and guidance on the selection and development of the Bayesian analysis method.

## Author contributions

**Conceptualization:** Johane H. Bracamonte, Jeffrey W. Holmes.

**Data curation:** Johane H. Bracamonte, Betty Pat, Louis J. Dell'Italia.

**Formal analysis:** Johane H. Bracamonte, Lionel Watkins.

**Funding acquisition:** Johane H. Bracamonte, Louis J. Dell'Italia, Jeffrey J. Saucerman, Jeffrey W. Holmes.

**Investigation:** Johane H. Bracamonte, Betty Pat, Louis J. Dell'Italia, Jeffrey J. Saucerman, Jeffrey W. Holmes.

**Methodology:** Johane H. Bracamonte, Jeffrey J. Saucerman, Jeffrey W. Holmes.

**Project administration:** Jeffrey W. Holmes.

**Resources:** Betty Pat, Louis J. Dell'Italia, Jeffrey J. Saucerman, Jeffrey W. Holmes.

**Software:** Johane H. Bracamonte, Lionel Watkins, Jeffrey J. Saucerman.

**Supervision:** Johane H. Bracamonte, Jeffrey W. Holmes.

**Validation:** Johane H. Bracamonte, Jeffrey W. Holmes.

**Visualization:** Johane H. Bracamonte, Lionel Watkins.

**Writing – original draft:** Johane H. Bracamonte, Jeffrey W. Holmes.

**Writing – review & editing:** Johane H. Bracamonte, Louis J. Dell'Italia, Jeffrey J. Saucerman, Jeffrey W. Holmes.

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
