## [Decision Letter · Decision Letter 0]

6 Sep 2024

Dear Dr. Holmes,

Thank you very much for submitting your manuscript "Contributions of mechanical loading and hormonal changes to eccentric hypertrophy during volume overload: a Bayesian analysis using logic-based network models." for consideration at PLOS Computational Biology.

As with all papers reviewed by the journal, your manuscript was reviewed by members of the editorial board and by several independent reviewers. In light of the reviews (below this email), we would like to invite the resubmission of a significantly-revised version that takes into account the reviewers' comments.

I am submitting the decision to request major revisions because I believe that the methodology developed here is novel and powerful and the application is highly innovative. That said, please note that both reviewers raised substantial concerns, with one reviewer recommending rejection. Addressing these reviewer concerns will greatly improve the potential impact of your work.

We cannot make any decision about publication until we have seen the revised manuscript and your response to the reviewers' comments. Your revised manuscript is also likely to be sent to reviewers for further evaluation.

Sincerely,

Daniel A Beard

Section Editor

PLOS Computational Biology

Daniel Beard

Section Editor

PLOS Computational Biology

Reviewer's Responses to Questions

**Comments to the Authors:**

Reviewer #1: The manuscript from Bracamonte et al. is very interesting and provides some notable findings:

First, it highlights the important role of Endothelin1 receptor activity in driving hypertrophic responses.

Second, it provides a plausible explanation for the suppression of mechano-transduction signaling in experimental volume overload (VO).

This challenges the common assumption that VO hypertrophy is driven by myocyte overstretch, as used in most computational models.

Additionally, though less supported by results and data, the authors claim that the model can replicate drug responses not used in its calibration and predicts that combining ERA with ARB may effectively reduce cardiomyocyte hypertrophy and dysfunction in VO.

Overall, the paper offers significant insights, especially for the computational modeling community.

However, several concerns must be addressed before it can be considered for publication in PLOS Computational Biology

1) While the abstract and introduction are clear and well-written, the presentation of the methods and results in the manuscript is somewhat sloppy, making it difficult to follow and understand. There are multiple instances where figure labels are incorrect, variable names in equations are inconsistent, and reference citations are either inconsistent or missing from the References list. The manuscript sometimes feels like an early draft and will require thorough proofreading and revision in this regard.

For example (a partial list of such issues):

*) Lines 148, 158: LVM/BM or LVM/BW

*) Line 204: is i an index for a given time?

*) Line 214: what is myoStrain?

*) CellArea is introduced in l 288, Is "Cell Area" above on page 18, l 271,276 the same?

*) l346 "and" missing

*) l388: should be 0.05 < w < 0.06, at least according to S1.5

*) Figure 4 and 5 seem to be switched

*) l 413: I don't really see that from Figure S1.11. Both network representations have the label S1.11 in the supplement.

*) l 428: "... of ANP, BNP, and bMHC (Fig 5)." -> Figure 6?

*) l.476 Delley et al. missing reference number in [] (This is the case for many references in the manuscript).

Griffin et al. is missing in the reference list.

*) l.637, reference numbers missing; I don't see a Dell'Italia 1995 paper

*) From S6 to S7: are the superscripts removed here on purpose?

*) S8, S9: not all the variables used in these Equations are introduced.

*) "to estimate the likelihood of the solution (Figure S1.1b)." should be S1.2b

*) S10: "The mechanical strain is to the network input myoStrain with an exponential function:"

A word ("mapped"?) is missing.

2) The strain computation outlined in the manuscript is not very clear.

In particular Lines 214-215 and Eq (2) have to be improved. Where does this come from, what are the different variables?

The supplemental material helps a bit but chapter S1.3 should be revised as well.

Figure S1.3: " [...] estimation of stretch/stress probability" stretch/strain?

3) "The PDF display the expected trends over time, that is, an acute increase in strain owing to the sudden increase of Ved, followed by a gradual decrease driven by the compensatory hypertrophic response (Fig 3)." I somehow struggle to see that behavior in Figure 3. I see an increase in Ved in 3b, maybe not a "sudden increase" and strain starts at approx. 1.2 in 3c) Is this the acute increase? If so, the figure should be improved to make this clearer.

4) The validation is not very convincing, in particular Fig 7c, CellArea fold change. Most data points are outside the 67% confidence interval and validating a model built on rats and dogs with data from mice is not optimal. Maybe there are more experimental data to strengthen this part of the study.

5) Furthermore, the model's ability to replicate drug responses and thus its predictive quality are only partially supported by the data and results. The changes shown in Figure 8a differ significantly in magnitude from the experimental data. In Figure 8b, it is unclear whether the changes in VO+ARB are statistically significant given the error bars. Additionally, in Figure 8c, the model fails to replicate the significant increase observed at t/tau = 2.7.

Minor:

*) for the sake of completeness (and its importance for the study) ET-1 should also be introduced around line 98

Reviewer #2: The authors combine data from 70 studies "on experimental volume overload (VO) in dogs and rats. This paper extracts data from these various papers to create a statistical models using data extracted from various papers. The authors claim the model "reproduces a number of responses to drug therapy not used in its calibration, and predicts that a combination of endothelin receptor antagonist and angiotensin receptor blockers would have the greatest potential to dampen cardiomyocyte hypertrophy and dysfunction in VO".

Assessment

Strengths

Mitral regurgitation continues to be a significant clinical problem, despite the recent advances in valve repair and replacement. So there might be some interest in this study.

Weaknesses

With effective valve replacement procedures in place, interest in treating cardiac disease associated with mitral regurgitation with various medications is of limited interest.

There are many studies included. To make any sense of data from different studies it is, in my opinion, imperative that the conditions between the studies be comparable (time course, disease severity, age of animals, gender etc etc) and carefully considered. There is insufficient information and discussion provided by the authors demonstrating how and whether the studies were comparable.

There are many ways to introduce volume overload and the severity depends on many factors. There is no discussion in the paper on how issues related to model diversity and complexity were addressed. Without careful consideration for technical differences between the various studies, it is difficult to understand how anything meaningful can come from simply pooling data from these many studies.

The method for estimating the myocardial strain is unlikely to be valid. In the end, strain must be measured dynamically and using interventions that monitor chamber or muscle deformation as a function of an applied changes in force or pressure. I have no confidence that the method used is in this paper is meaningful.

With respects to the biochemical measurements, it is not clear how many studies made measurements for the various factors incorporated into the model (such as adrenaline, Ang II, ANP etc etc). There was also no discussion or consideration for how and when these measurements were made in relationship to the time course of the volume overload condition (i.e. stage of the disease).

**Have the authors made all data and (if applicable) computational code underlying the findings in their manuscript fully available?**

Reviewer #1: Yes

Reviewer #2: **No: ** I did not see a reference to this

PLOS authors have the option to publish the peer review history of their article (what does this mean? ). If published, this will include your full peer review and any attached files.

**Do you want your identity to be public for this peer review?** For information about this choice, including consent withdrawal, please see our Privacy Policy .

Reviewer #1: No

Reviewer #2: No
---

## [Decision Letter · Decision Letter 1]

23 Feb 2025

Dear Dr. Holmes,

We are pleased to inform you that your manuscript 'Contributions of mechanical loading and hormonal changes to eccentric hypertrophy during volume overload: a Bayesian analysis using logic-based network models.' has been provisionally accepted for publication in PLOS Computational Biology.

Best regards,

Daniel A Beard

Section Editor

PLOS Computational Biology

Daniel Beard

Section Editor

PLOS Computational Biology

Reviewer's Responses to Questions

**Comments to the Authors:**

Reviewer #1: The authors thoroughly addressed my concerns, providing well-reasoned and scientifically robust responses to my feedback.

Reviewer #2: I am satisfied by the responses to my previous review.

**Have the authors made all data and (if applicable) computational code underlying the findings in their manuscript fully available?**

Reviewer #1: Yes

Reviewer #2: Yes

PLOS authors have the option to publish the peer review history of their article (what does this mean? ). If published, this will include your full peer review and any attached files.

**Do you want your identity to be public for this peer review?** For information about this choice, including consent withdrawal, please see our Privacy Policy .

Reviewer #1: No

Reviewer #2: **Yes: ** Peter H Backx

---

## [Editor Report · Acceptance letter]

PCOMPBIOL-D-24-01322R1

Contributions of mechanical loading and hormonal changes to eccentric hypertrophy during volume overload: a Bayesian analysis using logic-based network models.

Dear Dr Holmes,

I am pleased to inform you that your manuscript has been formally accepted for publication in PLOS Computational Biology. Your manuscript is now with our production department and you will be notified of the publication date in due course.

With kind regards,

Zsuzsanna Gémesi
